# Determinants of condom use during last sexual intercourse among male college youth of Kaski, Nepal: A cross-sectional survey

**Bijaya Parajuli, Chiranjivi Adhikari** *, **Narayan Tripathi**

School of Health and Allied Sciences, Pokhara University, Kaski, Nepal

* chiranadhikari@gmail.com

## Abstract

### Background

The National Family Planning program of Nepal has introduced the condom as an important family planning method. Despite the continuous effort from the public and private sectors at various levels, its use among youth remains low. Therefore, this study aimed to assess the factors associated with condom use during the last sexual intercourse among male college youth.

### Methods

We conducted a cross-sectional study and analyzed the responses of 361 male college youth (aged 19 to 24 years who reported being sexually active preceding six months of the survey), among the 903 participants who reported being involved in vaginal and anal sexual intercourse. The chi-square test was primarily used to find the associated factors and then, stepwise logistic regression was performed by selecting the covariates after the multicollinearity test followed by adjustment of confounders.

### Results

We found that more than one-fourth (27.4%) of the sexually active male youth had used the condoms during their last sexual intercourse. Postgraduate male youth were four times more likely to use the condoms during the last sexual intercourse than undergraduate male youth (AOR = 4.09, 95% CI; 2.08–8.06). Similarly, married youth were less likely to use the condoms during the last sexual intercourse with 95% lower odds than their counterparts (AOR = 0.05, 95% CI; 0.01–0.38). Male youth with adequate knowledge about the condoms were 8 times more likely to use them compared to those with inadequate knowledge (AOR = 8.42, 95% CI; 4.34–16.33). Likewise, male youth with favorable attitude towards the condoms were 2.5 times more likely to use them compared to their counterparts (AOR = 2.58, 95% CI; 1.23–5.42). Similarly, male youth having two or more sex partners were 4.5 times more likely to use the condoms than the youth having only a sex partner (AOR = 4.57, 95% CI; 2.38–8.76).

**Data Availability Statement:** The data are available in The Supporting information files.

**Funding:** The authors received no specific funding for this work.

**Competing interests:** The authors have declared that no competing interests exist.

## Conclusion

The study concluded that slightly more than one-fourth (27.4%) of male college youth in Kaski district used the condoms during their last sexual intercourse. Level of education, marital status, knowledge about condoms, attitude toward condoms, and number of sex partners are the determinants of condom use among male college youth so recommended for early behavioral interventions, especially in knowledge and attitude. Further studies focusing on including the rural youth and larger geography may help to reach a firmer conclusion.

## Introduction

Adolescents are defined as the people belonging to the age between 10 to 19 years while young and youth fall under 10–24 and 15–24 age groups respectively [1]. Globally, 1.2 billion youth account for one out of every six people worldwide [2]. Nearly, every fifth (19.38%) Nepalese is a youth, and so, youth's sexual and reproductive health is an important scope for Nepal, which is also prioritized, especially the youth and urbanization, in the official documents of the government [3].

The youth face various health problems that can affect their quality of life. First and foremost is HIV/AIDS, which is increasingly affecting young people [4]. There is variation in the sexual activities of youth between the countries of the world [5]. However, a common pattern is for youth reaching puberty and engaging in sexual activity earlier [6]. Evidence shows that youth with sexual initiation in their early life are more likely to have multiple sex partners and are at risk of sexually transmitted infections [7]. The previous study conducted in Kaski, Nepal showed that nearly two-thirds (60.4%) of the sexually active male youth had multiple sex partners [8]. Also, a study conducted in Achham district of Nepal showed that the prevalence of condom use among sexually active youth during the last two sexual intercourses was 31% and 35% [9]. Similarly, less than half of the total youth students in Kathmandu valley of Nepal had only used the condoms while engaging in sexual intercourse with commercial sex workers [10]. Low prevalence of the condom use among sexually active youth puts them at risk of sexually transmitted infections, unsafe abortion, unwanted pregnancies, and pregnancy-related complications [11]. According to the Nepal Demographic Health Survey 2016, youth aged 15 to 24 years were more likely to report STI and its symptoms than older men. Relatedly, just more than two-thirds (68.9%) of the same age group had used the condoms during last sexual intercourse with multiple sex partners and 56.2% had used them during the last paid sexual intercourse [12]. In line with, the recent data from the National Centre for AIDS and STI control show that the prevalence of HIV infections among the age group of 20–24 years in Nepal was 11.5% [13]. Thus, the literature suggests that the sexual activity of youth in Nepal is risky.

The National Family Planning program of Nepal introduced the condom as an important family planning method. Moreover, it is also being promoted through social marketing in Nepal by various organizations. Existing policies and programs and their focus are on condom use for sexually active youth and its easy access throughout the country [12]. However, the socio-cultural context of Nepal imposes barriers to using it among unmarried youth as sexual activities are not easily acceptable among them in Nepal [14]. A previous study among male college youth of Kaski district revealed that a substantial proportion of them was engaged in risky sexual behavior where more than half (51.61%) of them had not used the condoms at every act of sexual intercourse with commercial sex worker [8]. Moreover, studies have also

documented that living with parents and higher education levels are important predictors of consistent condom use [15]. Similarly, knowledge and attitude toward condom use can greatly impact condom use [16]. Thereupon, this study, along with prevalence, also aimed to examine these factors among male college youth in Kaski, Nepal.

## Materials and methods

### Study setting and design

A cross-sectional survey was conducted among male college students aged 19–24 years from 25 bachelor's and master's level colleges in Kaski district of Nepal. To calculate the sample size, the prevalence of condom use during the last sexual intercourse among sexually active youth was taken from a study conducted in Kathmandu which was 39% [17]. With permitted error (5%), Z-value (1.96), finite population correction (FPC for total students of 6369), non-response rate (23%) that was taken from a similar study conducted in Nepal [9], and a design effect of two, we obtained 849. However, we reached out to 903 sexually active youth from whom only 361 were involved in vaginal or anal sexual intercourse and were enrolled in the study.

The list of the bachelor and master level colleges of Kaski district, Nepal was obtained from the Rural municipality/municipality profile of Kaski, 2018. Then, the number of male youth of each class of the colleges was assessed through the information provided by related colleges. The total number of classrooms of undergraduate level as bachelor first, second, third and fourth year were 87, 86, 84 and 58 respectively and the postgraduate level was 21. Based on the average classroom size, 46 classes (Bachelor first year = 12, second year = 12, third year = 11, fourth year = 8, and Masters first year = 3, proportionately based on the population) were randomly selected for each grade to get the sample size. Thus, male college youth of the selected classes were the study sample.

### Participants

The participants were sexually active male college youth aged 19–24 years, who reported their vaginal or anal sexual intercourse within six months preceding the survey. However, we excluded the incomplete responses and those not meeting the criteria from the analysis.

### Data collection

Data were collected through a self-administered questionnaire in the selected bachelor and master level colleges of Kaski, Nepal in 2019. Two male enumerators were recruited and trained with a two days' workshop for data collection procedures and ethical considerations.

Before data collection, a brief script was prepared to maintain the consistency of the information provided to each class. An orientation session of 10–15 minutes was conducted in each selected class by male enumerators. During orientation, students were made clear about the purpose of the study and also, praised for their participation. They were made assured of the anonymity and the benefits and risks. After clearing the students' curiosity, verbal consent was obtained. The students were requested to open and fill out the sealed questionnaire and then submit it to the enumerator in the same envelope.

### Questionnaire

An Illustrative questionnaire for an interview-survey with young people was used to assess the participants' knowledge about the condoms and their use and attitude toward them [18].

Similarly, to assess the behavioral factors, Nepal Adolescent and Youth Survey 2011/12 questionnaire was used [19].

## Measures

**Condom use during last sexual intercourse.**   The outcome of the study was condom use during the last sexual intercourse. For the study purpose, last sexual intercourse refers to the latest vaginal or anal sexual intercourse that was carried out within six months preceding the survey. It was assessed with the question, 'Did you or your partner use any contraceptive devices during your latest sexual intercourse? The respondents answered this question with two options, yes and no. Those who answered yes were asked the question, 'Which contraceptive device did you or your partner use during your latest sexual intercourse? The options for this question were contraceptive pills, contraceptive injections, emergency contraceptive pills, and the condoms. Those who did not use any contraceptive devices were asked about alternative measures. The options were: withdrawal and safe period. For the analysis purpose, only those who answered condom during their latest sexual intercourse were considered condom users during the latest sexual intercourse.

**Indicators of socio-demographic variables.**   Age, marital status, place of permanent residence, and living arrangement of the respondents were used as sociodemographic variables.

**Condom knowledge.**   Condom knowledge was assessed with five items (Score, 0–5; a median, 3) which was adopted from the illustrative questionnaire for young people, which was further dummied as inadequate; median value below 3 and adequate; $\geq$ 3. The items were related to the condom that protects against sexually transmitted infections including HIV and AIDS, pregnancy, and its use. The internal consistency (Cronbach's alpha) of the scale was 0.56. The mean covariance between the items was 0.08 and the item mean-variance was 1.95. For construct validity, principal component analysis was conducted which revealed one factor with eigenvalues greater than 1 and the total variance explained was 47.31. The content validity of the scale was established using a panel of three judges competent in the field of reproductive health, who were requested to assess the relevance of the content used in the questionnaire.

**Attitude toward condom.**   Participants' attitude toward condom was examined with eight items (Score, 0–8; a median, 5), which was further dummied as unfavorable attitude; median value less than 5 and favorable attitude; median $\geq$5. The internal consistency (Cronbach's alpha) of the scale was 0.73. The mean covariance between the items was 0.18 and the item-mean variance was 1.50. The construct and content validity of the attitude scale was also conducted using the same steps of the knowledge scale. Principal component analysis for assessing construct validity revealed two factors with eigenvalues greater than 1 and the total variance explained was 49.32 and 23.96% respectively.

**Sexual behavior.**   Three sexual behaviors of the participants were measured: age at sexual debut, number of sex partners, and relationship with the sex partner. Age at sexual debut and the number of sex partners were assessed as continuous variables. However, they were later categorized for analysis purposes.

## Statistical methods

Data were entered into the Epi Data 3.1 version and exported to SPSS for analysis. Descriptive statistics were applied to describe the characteristics of the study population and the prevalence of condom use during the last sexual intercourse. Chi-square ($\chi$2) statistics were used to identify the association between explanatory variables and condom use during the last sexual intercourse. Stepwise logistic regression was performed by selecting the covariates after the multicollinearity test followed by adjustment of confounders. The variables were considered

significant at a p-value less than 0.05. Model 1 included the socio-demographic variables; model 2 comprised the factors of model 1 and knowledge variable; model 3 included the factors of model 2 and attitude variable; and model 4 comprised the factors of model 3 and behavioral variables. Altogether there were 74 ambiguous responses of the sexually active male youth in which 12 responses were of sexually active male youth within six months preceding the survey. Likewise, among 21 incomplete responses, 6 largely incomplete responses were of sexually active male youth within six months preceding the survey. Thus, ambiguous responses that were logically inconsistent were excluded after verification through the consultation of reproductive health experts and incomplete data were also eliminated from the analysis.

### Ethical considerations

The ethical process was approved by the Institutional Review Committee of Pokhara University. Also, a letter of permission was obtained from the Ministry of Social Development, Gandaki Province. Respondents were assured and maintained confidentiality and anonymity. Oral consent was obtained from the participants and no personal identifiers were recorded. They were provided information about the voluntary nature of participation in the survey and the provision of withdrawal from the survey at any time.

## Results

Altogether 27 students were not present on the day of survey administration whereas 41 students decided not to participate before the start of the survey or chose to stop participation before completing the entire survey. Another 74 ambiguous responses and 21 incomplete responses were eliminated from the analysis. The valid responses for the analyses were 903 sexually active male college youth. However, only 361 were found to be involved in vaginal or anal sexual intercourse within six months before the study and henceforth analyzed.

Table 1 shows that nearly two-thirds (64.81%) of the respondents belong to the age group of 19–21 years. The mean age of the respondents was 20.97±1.56 years. Similarly, more than

**Table 1. Socio-demographic variables (n = 361).**

| Variables | Frequency | Percentage (%) |
|---|---|---|
| Age of the respondents (Range = 19–24, Mean ±Standard Deviation 20.97±1.56) | | |
| 19–21 years | 234 | 64.81 |
| 22–24 years | 127 | 35.18 |
| Level of education | | |
| Undergraduate | 243 | 67.31 |
| Postgraduate | 118 | 32.68 |
| Permanent residence | | |
| Kaski | 138 | 38.22 |
| Outside Kaski | 223 | 61.77 |
| Marital status | | |
| Married | 37 | 10.24 |
| Unmarried | 324 | 89.75 |
| Currently living with | | |
| With family | 173 | 47.90 |
| With relatives | 34 | 2.50 |
| With friends | 9 | 9.40 |
| Alone | 145 | 40.20 |

Table 2. Condom knowledge of college youth (n = 361).

| Statements | In group n (%) | % used condom | p-value |
|---|---|---|---|
| **1.Condoms are an effective method of preventing pregnancy** | | | |
| Agree | 217(60.1) | 48.9 | <0.001 |
| Don't know | 2 (0.6) | 0.8 | |
| Disagree | 142 (39.3) | 50.4 | |
| **2.Condoms can be used more than once** | | | |
| Agree | 21 (5.8) | 7.3 | 0.015 |
| Don't know | 8 (2.2) | 2.7 | |
| Disagree | 332 (92.0) | 90.1 | |
| **3.Condoms are an effective way of protecting against HIV/AIDS** | | | |
| Agree | 257 (71.2) | 62.2 | <0.001 |
| Don't know | 9 (2.5) | 3.4 | |
| Disagree | 95 (26.3) | 34.4 | |
| **4.Condoms can slip off the man and disappear inside the woman's body** | | | |
| Agree | 20 (5.5) | 4.6 | 0.048 |
| Don't know | 24 (6.6) | 8.4 | |
| Disagree | 317 (87.8) | 87.0 | |
| **5.Condoms are an effective way of protecting against sexually transmitted diseases** | | | |
| Agree | 243 (67.3) | 58.8 | <0.001 |
| Don't know | 4 (1.1) | 1.5 | |
| Disagree | 114 (31.6) | 39.7 | |

two–thirds (67.31%) of the respondents were undergraduate students. Nearly two out of three (61.77%) of the total respondents from the study were not the permanent residents of Kaski. Similarly, the majority (89.75%) of the respondents were unmarried. Nearly half (40.2%) of the respondents lived alone.

For the 'agree' option 1 score was given while 'disagree' and 'don't know' scored 0 and summed up for knowledge (Table 2) and attitude (Table 3). A score below the median of the condom knowledge was classified as inadequate, and the median and above as an adequate. Similar scoring procedure was applied for the condom attitude as unfavorable and favorable.

Table 4 demonstrates that more than half of the respondents (53.2%) had inadequate knowledge about the condom and nearly three-fifths (56%) had an unfavorable attitude toward the condom.

Table 5 shows that nearly half (47.5%) of the respondents had their first sexual intercourse at the age below 18 years. Nearly two-thirds (60.4%) of the sexually active respondents had two or more than two partners. More than three fourth (81.7%) of the total respondents had a girl-friend as their latest sex partner.

Condom use during the last sexual intercourse was the main outcome of the study. As shown in Table 5 nearly three-fourths (72.6%) of the sexually active male youth who had reported sexual intercourse within six months preceding the survey, did not use a condom during their last sexual intercourse. Nearly half (46.94%) of the condom non-users had used alternative measures. Similarly, nearly one-third (30.89%) of the condom non-users had used withdrawal followed by a safe period (26.83%), emergency contraceptive pills (19.51%), contraceptive injection (13%), contraceptive pills (9.76%), and abortion (4.88%).

**Table 3. Attitude towards condom (n = 361).**

| Statements | In group n (%) | % used condom | p-value |
|---|---|---|---|
| **1.A girl can suggest to her boyfriend to use a condom** | | | |
| **Agree** | 205(56.8) | 45.0 | <0.001 |
| **Don't know** | 7(1.9) | 2.7 | |
| **Disagree** | 149(41.3) | 52.3 | |
| **2.A boy can suggest to his girlfriend to use a condom** | | | |
| **Agree** | 198 (54.8) | 42.4 | <0.001 |
| **Don't know** | 8(2.2) | 3.1 | |
| **Disagree** | 155(42.9) | 54.6 | |
| **3.Condoms are suitable for casual relationships** | | | |
| **Agree** | 175(48.5) | 34.0 | <0.001 |
| **Don't know** | 7 (1.9) | 2.7 | |
| **Disagree** | 179 (49.6) | 63.4 | |
| **4.Condoms are suitable for steady, loving relationships** | | | |
| **Agree** | 140 (38.8) | 29.0 | <0.001 |
| **Don't know** | 7 (1.9) | 2.7 | |
| **Disagree** | 214 (59.3) | 68.3 | |
| **5.It would be too embarrassing for someone like me to buy or obtain condom** | | | |
| **Agree** | 230 (63.7) | 69.8 | <0.001 |
| **Don't know** | 4 (1.1) | 1.1 | |
| **Disagree** | 127 (35.2) | 29.0 | |
| **6.If a girl suggested using condom to her partner, it would mean that she didn't trust him** | | | |
| **Agree** | 190 (52.6) | 65.6 | <0.001 |
| **Don't know** | 8(2.2) | 2.7 | |
| **Disagree** | 163 (45.2) | 31.7 | |
| **7.Condom reduces sexual pleasure** | | | |
| **Agree** | 278 (77) | 80.9 | 0.002 |
| **Don't know** | 10(2.8) | 3.4 | |
| **Disagree** | 73 (20.2) | 15.6 | |
| **8.If unmarried couples want to have sexual intercourse before marriage, they should use the condom** | | | |
| **Agree** | 184 (51) | 37.8 | <0.001 |
| **Don't know** | 8 (2.2) | 3.1 | |
| **Disagree** | 169(46.8) | 59.2 | |

## Bivariate analysis

We found that age of the respondents ($\chi^2$ = 29.08, p-value <0.001), level of education ($\chi^2$ = 23.70, p-value <0.001), permanent residence ($\chi^2$ = 10.20 p-value = 0.001),marital status

**Table 4. Knowledge and attitude category (n = 361).**

| Category | Frequency | Percentage (%) |
|---|---|---|
| **Condom knowledge (Median,3; Min-Max, 0–5)** | | |
| **Inadequate knowledge (<3)** | 192 | 53.2 |
| **Adequate knowledge ($\geq$ 3)** | 169 | 46.8 |
| **Condom attitudes (Median,5; Min-Max, 0–8)** | | |
| **Unfavorable attitude (<5)** | 202 | 56.0 |
| **Favorable attitude ($\geq$5)** | 159 | 44.0 |

**Table 5. Behavioral factors (n = 361).**

| Variables | Frequency | Percentage (%) |
|---|---|---|
| **Age at first sexual intercourse (Mean ±SD, 18.09±2.05; Min-Max,12–23)** | | |
| **<18 years** | 171 | 47.5 |
| **≥18 years** | 190 | 52.8 |
| **Number of sex partner** | | |
| **One** | 143 | 39.6 |
| **≥2** | 218 | 60.4 |
| **Latest sex partner** | | |
| **Wife** | 33 | 9.14 |
| **Girlfriend** | 295 | 81.7 |
| **Casual friend** | 13 | 3.6 |
| **Sex worker** | 20 | 5.5 |
| **Condom use during last sexual intercourse** | | |
| **Yes** | 99 | 27.4 |
| **No** | 262 | 72.6 |
| **Alternative of condom among condom non-users (n = 262)** | | |
| **Yes** | 123 | 46.9 |
| **No** | 139 | 53.1 |
| **Alternative measures used by condom non-users (n = 123)** | | |
| **Safe period** | 33 | 26.8 |
| **Withdrawal** | 38 | 30.9 |
| **Contraceptive pills** | 12 | 9.8 |
| **Contraceptive injection** | 16 | 13.0 |
| **Emergency contraceptive pills** | 24 | 19.5 |

($\chi^2$ = 12.65, p-value <0.001), living arrangement ($\chi^2$ = 15.28, p-value <0.001), knowledge about condom ($\chi^2$ = 62.15,p-value <0.001),attitude towards condom ($\chi^2$ = 80.77, p-value <0.001) and number of sex partners ($\chi^2$ = 74.50, p-value <0.001) significantly associated with condom use during the last sexual intercourse. There were no significant differences between age at first sexual intercourse, relationship with sex partner and condom use during the last sexual intercourse (Table 6).

## Multivariate analysis

In the model (Table 7), adjusted odds ratio (AOR) at 95% Confidence Interval was calculated from logistic regression analysis in order to examine the association between selected variables and condom use during the last sexual intercourse. The variables were considered significant at a p-value less than 0.05. Altogether four models were developed where in the first model variables of socio-demographic characteristics were incorporated. Knowledge, attitude, and behavioral factors were added in the second, third, and fourth models, respectively. After assessing multicollinearity in the variables, it was found that the age of the respondent and level of education were highly correlated (r = 0.9). Thus, the variable 'respondent's age' was not entered in the logistic model. Among the eight variables that were incorporated in the model, five variables remained statistically significant in the final model after controlling for other variables.

As shown in Table 7, postgraduate male youth were 4 times more likely to use a condom during last sexual intercourse compared to undergraduate male youth (AOR = 4.09, 95% CI; 2.08–8.06). Similarly, married male youth were less likely to use a condom during last sexual

**Table 6. Association of different variables with condom use during the last sexual intercourse among the youth (n = 361).**

| Variables | Condom used | | Pearson chi-square ($\chi^2$) | p-value | UOR (At 95% CI) |
|---|---|---|---|---|---|
| | Yes | No | | | |
| **Age of the respondents** | | | | | |
| **19–21 years** | 148(63.2) | 86(36.8) | 29.08 | <0.001* | 0.19(0.10–0.36) |
| **22–24 years** | 114(89.8) | 13 (10.2) | | | Ref |
| **Level of education** | | | | | |
| **Undergraduate** | 157(64.6) | 86 (35.4) | 23.70 | <0.001* | 4.42(2.34–2.34) |
| **Postgraduate** | 105(89) | 13(11) | | | Ref |
| **Permanent residence** | | | | | |
| **Kaski** | 51(36.95) | 87(63.04) | 10.20 | 0.001* | 2.13(1.33–3.42) |
| **Outside Kaski** | 48(21.52) | 175(78.47) | | | Ref |
| **Marital status** | | | | | |
| **Married** | 1(2.70) | 36(97.29) | 12.65 | <0.001*# | 0.06(0.009–0.47) |
| **Unmarried** | 98(30.24) | 226(69.75) | | | Ref |
| **Living arrangement** | | | | | |
| **With family** | 64(37) | 109(63.01) | 15.28 | <0.001* | 2.56(1.58–4.14) |
| Alone and with others [a] | 35(18.61) | 153(81.38) | | | Ref |
| **Knowledge about condom** | | | | | |
| **Inadequate knowledge** | 106(55.2) | 86 (44.8) | 62.15 | <0.001* | Ref |
| **Adequate knowledge** | 156(92.3) | 13(7.7) | | | 9.73(5.16–18.34) |
| **Attitude towards condom** | | | | | |
| **Unfavorable attitude** | 11(6.14) | 168(93.85) | 80.77 | <0.001* | 0.070(0.036–0.13) |
| **Favorable attitude** | 88(48.35) | 94(51.64) | | | Ref |
| **Age at first sexual intercourse** | | | | | |
| **<18 years** | 45(26.31) | 126(73.68) | 0.207 | 0.649 | 0.89(0.56–1.43) |
| **≥18 years** | 54(28.42) | 136(71.57) | | | Ref |
| **Number of sex partners** | | | | | |
| **1** | 68(47.55) | 75(52.44) | 74.50 | <0.001* | Ref |
| **≥2** | 194(89) | 24(11) | | | 8.91(5.21–15.24) |
| **Relationship with the sex partner** | | | | | |
| **Girlfriend and wife** | 233(71.9) | 91(28.1) | 0.69 | 0.26 | 1.41(0.62–3.21) |
| **Others[b]** | 29(78.4) | 8(21.6) | | | Ref |

[a]With friends and relatives,

[b]Casual friend and sex worker,

[#] p-value from Fisher Exact;

Statistically significant at

*p<0.05

intercourse with 95% of lower odds than unmarried male youth (AOR = 0.05, 95% CI; 0.01–0.38).

The variables that were significant in the first model retained significance in the second model even after adjusting the knowledge variable. Model 2 showed that male youth having adequate knowledge about the condoms were 8 times more likely to use them during last sexual intercourse than male youth having inadequate knowledge about them (AOR = 8.42, 95% CI; 4.34–16.33). Model three explained that youth having the favorable attitude toward the condoms were 2.5 times more likely to use them during their last sexual intercourse compared to those having unfavorable attitudes (AOR = 2.58, 95% CI; 1.23–5.42) (Table 7).

**Table 7. Adjusted odds ratio to examine the strength of association between condom use during last sexual intercourse and related variable (n = 361).**

| Variable | Model I | Model II | Model III | Model IV |
|---|---|---|---|---|
| Level of education | | | | |
| Undergraduate | Ref | Ref | Ref | Ref |
| Postgraduate | 4.09(2.08–8.06)** | 3.87(1.87–7.99)** | 3.61(1.74–7.50)** | 3.58(1.45–8.81)** |
| Marital status | | | | |
| Married | 0.05(0.01–0.38)** | 0.06(0.01–0.47)** | 0.06(0.01–0.50)** | 0.07(0.01–0.57)** |
| Unmarried | Ref | Ref | Ref | Ref |
| Permanent residence | | | | |
| Kaski | 1.33(0.65–2.72) | 1.31(0.60–2.89) | 1.40(0.62–3.11) | 1.26(0.52–3.06) |
| Outside Kaski | Ref | Ref | Ref | Ref |
| Living arrangement | | | | |
| With family | 3.11(1.53–6.29)** | 2.32(1.07–5.03)** | 1.99(0.90–4.41) | 2.03(0.84–4.89) |
| With others [a] | Ref | Ref | Ref | Ref |
| Knowledge about condom | | | | |
| Inadequate | | Ref | Ref | Ref |
| Adequate | | 8.42(4.34–16.33)** | 5.31(2.53–11.11)** | 4.14(1.81–9.46)** |
| Attitude toward condom | | | | |
| Unfavorable | | | Ref | Ref |
| Favorable | | | 2.58(1.23–5.42)** | 2.28(1.01–5.17)** |
| Age at sexual debut | | | | |
| <18 years | | | | 0.56(0.28–1.11) |
| ≥18 years | | | | Ref |
| Number of sex partners in life time | | | | |
| 1 | | | | Ref |
| ≥2 | | | | 4.57(2.38–8.76)** |
| Relationship with a latest sex partner | | | | |
| Girlfriend and wife | | | | 0.41(0.13–1.25) |
| Others [b] | | | | Ref |

[a]With friends and relatives,

[b]Casual friend and sex worker;

Statistically significant at

**p<0.05from multivariate analysis

Model 1: Background variables, Model 2: Background and knowledge variables, Model 3: Background, knowledge and attitude variables, Model 4: Background, knowledge, attitude, and behavioral variables

In the fourth model the variables: level of education, marital status, condom knowledge and attitude remained significant even after adjustment of the behavioral variable. Among the three variables of behavioral factor, the number of sex partners in lifetime remained statistically significant in model 4. The analysis indicates that male youth having two or more sex partners were 4.5 times more likely to use a condom during last sexual intercourse than male youth having one sex partner (AOR = 4.57, 95% CI; 2.38–8.76) (Table 7).

## Discussion

The study examined the determinants of condom use during the last sexual intercourse among sexually active male youth of Kaski, Nepal. The findings showed that more than half (55.98%)

male youth of Kaski, Nepal were sexually active whereas less than one-fourth (23%) of the total youth had experienced sexual intercourse within six months preceding the survey. Sexuality, although a sensitive issue and so the inconsistencies regarding self-reporting are rampant, this finding indicates that the majority of male youth in Kaski, Nepal are sexually active. The prevalence of sexual intercourse among youth in this study is higher than that of rural India where 30% of youth had sexual intercourse in the same age group of 19–24 years [20]. This difference may be due to rural-urban variation. However, the study's prevalence is lower than that of a similar study conducted among the youth of the USA, Tanzania, and Thailand, where prevalence was 57%, 70.06%, and 62.4% respectively [21–23]. This might be due to the differences in perception towards sexual intercourse and relationships. Also, one has to consider that methodological diversity may explain these differences.

This study revealed that less than one-third (27.4%) of male youth had used the condom during the last sexual intercourse which is consistent with the study conducted in Achham district of Nepal where condom use during the last sexual intercourse was 31% [9]. Despite the Nepal government's effort to make the condoms easily available and accessible, their use is low among male college youth. In countries such as Southern Brazil, China, Canada, United States, Ethiopia the prevalence of condom use in the last sexual intercourse among university students was 61.4%, 44.2%, 47.2%, 63.8%, and 55.8% respectively [24–28]. This difference might be due to the knowledge level about the condoms and youth's attitude to use them within and outside the country.

Consistent with the previous research [15], this study provides evidence that Postgraduate students were 4 times more likely to use a condom during the last sexual intercourse compared to undergraduate students (AOR = 4.09, 95% CI; 2.08–8.06). Similarly, more than half of the respondents had inadequate knowledge (53.2%) about condoms and their use. Male youth having adequate knowledge about condoms were 8 times more likely to use a condom during the last sexual intercourse than male youth having inadequate knowledge about condoms (AOR = 8.42, 95% CI; 4.34–16.33). The main reason for the gap in knowledge regarding condoms and their use among male college youth might be due to the socio-cultural environment in Nepal that imposes a barrier, especially among the unmarried youth to share knowledge regarding SRH [14]. In contrast to this finding, a study conducted in Ethiopia revealed that 75.1% of the male youth had adequate knowledge of condom use. The main reason for the high level of condom knowledge in Ethiopia may be due to the presence of the HIV/AIDS clubs in the schools [16] whereas this kind of provision is lacking in the Nepalese context.

The present study found that more than half (56.7%) of the total respondents had unfavorable attitudes toward the condom. In contrast to this finding, a study conducted in Ethiopia revealed that 83.6% had a favorable attitude toward condoms [16]. This difference is probably due to a fairly traditional Nepalese society that is changing slowly in terms of developing a more open attitudes toward sexual and reproductive health and its important aspects [14]. Multivariate analysis showed that the male youth having a favorable attitude toward the condoms were 2.5 times more likely to use them during the last sexual intercourse with compared to the male youth having an unfavorable attitude (AOR = 2.58, 95% CI; 1.23–5.42) which is consistent with the findings of Thailand, Croatia, United States and China [29–32]. Thus, attitude toward the condoms among youth is likely to develop the patterns of condom use behavior [33]. Also, it reflects the choice to use or ignore condoms among them [34].

In this study, more than three-fifths (60.4%) of the total youth had two or more sex partners, which is in line with the previous finding of a similar study conducted in the same age group of Kaski, Nepal [8]. However, this finding is quite more than the study conducted in the Achaam district of Nepal, where 28.2% of youth had two or more than two partners [9]. This inconsistency might be due to urban-rural variation. Also, permanent-temporary residence

variation might have fueled this gap as a previous study had shown that the youth in temporary residence were found to be involved in risky sexual behavior [10]. The minimum age at the sexual debut of male college youth in the study was 12 years so the curriculum addressing safe sex education at the school level should be provided before the initiation of the sexual activities.

The result of the study on the prevalence of condom use among married male youth was just one percent which is somewhat consistent with the finding of the Nepal Demographic Health Survey, in which it was 4% [12]. Also, married youth in this study were less likely to use the condoms during the last sexual intercourse with 95% of lower odds than unmarried youth (AOR = 0.05, 95% CI; 0.01–0.38). This seems particularly in line with previous studies indicating that cohabitation or being married was associated with low condom use [35]. Thus, this study provides additional evidence of differences in condom use according to the marital status which indicates the need to develop strategies responsive to address the diverse contexts of condom use as per marital status. It is important to note, however, that our study findings revealed less condom use among married youth but did not investigate the reasons for not using it. Therefore, the reasons for the non-use of condoms in marital relations need to be explored and addressed in Nepal.

This study has limitations to be noted. Because of the sensitive nature of the issue, the information from this study may be either under or over-reported although efforts were made to minimize the chance of reporting bias. Additionally, external validity may be limited to male college youth in Nepal. This study conducted in Kaski, Nepal is in the first row of its kind among unmarried youth that reflects the youth knowledge level of the condom, their attitude toward it, and its usage. Our findings extend previous research by identifying the factors associated with the condoms and alternative measures used by the youth. This study is of Kaski, Nepal but it reflects the scenario of sexual behavior of youth in Nepal because national-level studies focus mostly on married youth only. The findings of this study alert health program planners and policymakers including the non-governmental organizations to the popular belief of traditional Nepalese cultural norms where sexual relations happen only inside the married statuses and contraceptives are only important to married youth. Thus, more studies should be conducted to design the interventions for increasing the use of condoms among youth by using identified predictors of this study.

## Conclusion

The study concluded that slightly more than one-fourth (27.4%) of male college youth in Kaski district, Nepal had used a condom during their last sexual intercourse. Level of education, marital status, condom knowledge and attitude, and number of sex partners were the important determinants of condom use during last sexual intercourse among male college youth and recommended for early behavioral interventions. Further studies covering larger geography including rural areas may help to reach a firmer conclusion.

## Supporting information

**S1 Data. Data set.**
(SAV)

**S1 File. Questionnaire.**
(DOCX)

**S2 File. Bivariate analysis of all reported sexual active male youth.**
(DOCX)

**S3 File. Validation of the questionnaire.**
(DOCX)

**S4 File. Weighted & unweighted number.**
(DOCX)

## Acknowledgments

We are grateful to Dr. Shyam Thapa, for his guidance and technical support from protocol development to manuscript writing. We would like to thank Dr. Niranjan Shrestha, Assistant Professor, Pokhara University for his guidance on statistical aspects, and Mr. Chandra Prasad Khanal, for the professional language editing. We are thankful to all the participants who voluntarily provided their valuable information.

## Author Contributions

**Conceptualization:** Bijaya Parajuli, Chiranjivi Adhikari, Narayan Tripathi.

**Data curation:** Bijaya Parajuli.

**Formal analysis:** Bijaya Parajuli, Chiranjivi Adhikari.

**Investigation:** Narayan Tripathi.

**Methodology:** Bijaya Parajuli, Chiranjivi Adhikari, Narayan Tripathi.

**Project administration:** Bijaya Parajuli, Chiranjivi Adhikari, Narayan Tripathi.

**Software:** Bijaya Parajuli.

**Supervision:** Chiranjivi Adhikari.

**Validation:** Chiranjivi Adhikari.

**Visualization:** Bijaya Parajuli.

**Writing – original draft:** Bijaya Parajuli, Chiranjivi Adhikari, Narayan Tripathi.

**Writing – review & editing:** Bijaya Parajuli, Chiranjivi Adhikari.

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
