## [Decision Letter · Decision Letter 0]

18 Nov 2020

PONE-D-20-30105

Determinants of condom use during last sexual intercourse among male college youths of Kaski, Nepal: A cross-sectional survey

PLOS ONE

Dear Dr. Adhikari,

Thank you for submitting your manuscript to PLOS ONE. After careful consideration, we feel that it has merit but does not fully meet PLOS ONE’s publication criteria as it currently stands. Therefore, we invite you to submit a revised version of the manuscript that addresses the points raised during the review process.

There are a number of limitations in the present paper in terms of methodology that are described below. In addition we have found that a related study [https://www.jhas.org.np/jhas/index.php/jhas/article/view/134/114] has been published based on the same survey. You should be aware of PLOS ONE policy regarding related papers: If a submitted study replicates or is very similar to previous work, authors must provide a sound scientific rationale for the submitted work and clearly reference and discuss the existing literature. Submissions that replicate or are derivative of existing work will likely be rejected if authors do not provide adequate justification (http://www.plosone.org/static/publication#results). The mentioned paper was not included among the reference or put in context as required, and if the required justification and incorporation in the study is not given that is grounds for rejection. There are also concerns regarding the English language that needs to be copy edited before final acceptance.

Both reviewers pose a number of questions and make useful suggestions for the improvement of the paper. In particular there are concerns regarding the lack of a methods section, a lack of specified objectives, and limitations of the current analysis.

Regarding the lack of specified objectives and the setup, there is only a talk regarding “So this study examined the factors associated with condom use at last sexual intercourse among the male college youths”. What specific factors were in mind when carrying out the analysis? What is known regarding these factors in similar or other populations? There is talk regarding the DHS 2016 and levels of STI of youth. The DHS also recollected information on condom use at last sex and at last- paid sex. That information needs to be incorporated and discussed.

Regarding the methodological parts of the survey, the description is not enough as assessed by reviewer 1, including lack of data on the universe, the sampling procedure, protocol on reaching the units, patterns of missingness. The sample seems to have strata and a cluster design but apparently there are no weights present in the data. It does not seem that there are equal probabilities of selection given the research design, or adjustment for non-response. All this has to be justified / remedied.

Regarding the statistical analysis there is a tension between the two sets of factors that are being analyzed, some sociodemographic and others regarding attitudes towards condom use, which seem to be the most clearly related to condom use. The way they are addressed and combined is less than satisfactory:

4. As mentioned by both reviewers, the treatment of knowledge and attitude towards the use of condoms is not adequately explained. We don’t know whether there is any connection with this way of measuring and any of the previous literature. We don’t know why the variable has been defined in such an arbitrary way, first treating all items alike and then choosing a threshold. We don’t know the respective prevalence of the different items in the population, as mentioned by reviewer 2, or in the sample of sexually active individuals. You can check how scale validation can be carried out (and an example of a similar, but different scale, here https://www.ncbi.nlm.nih.gov/pmc/articles/PMC3608159/, see also 10.1037/0278-6133.13.3.224 on the UCLA condom attitude scale).5. As mentioned by reviewer 2, it would be good to extend the analysis of the knowledge and attitude items to the total population for comparison. I’d suggest presenting at least a tabulation of row percentages by item for the total population, the sexually active population, and among the latter, users and non-users. Item validity and reliability analysis can lead to an improved scale.6. Using both the attitude and knowledge and the sociodemographic variables in the same model to compute the AOR is not adequate. I’d suggest splitting current table 4 in 2, one with the tabulations and unadjusted AOR, and another one with model results. In the second one a model with only sociodemographic determinants should be reported, possibly a model only with the knowledge-attitude components, and if desired a model with all of them. For interpretation of sociodemographic patterns the first model should be used. The reason is that the model including attitude asks questions like “what is the probability of using condom according to living arrangement conditional on attitude and knowledge and the rest of SE factors”. That is not the question of interest. It is logical that use is connected to attitude towards condom use. We want to know what are the students that will be more likely to both use and have a more positive attitude towards it.7. Regarding the presentation of results and discussion, as both reviewers point out this section needs to be improved connecting the results to prior studies and noting the limitations of the current study (sample size among them). PLOS ONE endorses the use of the strobe statement that you should check in revising the introduction, the results section and the discussion, https://www.strobe-statement.org/fileadmin/Strobe/uploads/checklists/STROBE_checklist_v4_cross-sectional.pdf

There are in addition other minor points made by the reviewers that require attention.

We look forward to receiving your revised manuscript.

Kind regards,

José Antonio Ortega, Ph.D.

Academic Editor

PLOS ONE

Journal Requirements:

2. Please include additional information regarding the validation of the survey or questionnaire used in the study and ensure that you have provided sufficient details that others could replicate the analyses. Furthermore, if the questionnaire is not under a copyright more restrictive than CC-BY, please include a copy, in both the original language and English, as Supporting Information.

3. We noted in your submission details that a portion of your manuscript may have been presented or published elsewhere.

"The data for this manuscript is the subset of the data from which article published in Journal of Health and Allied Sciences (JHAS), however the research question/objectives, context/content, draft including others are different. I will attach the article already published in JHAS."

Please clarify whether this publication was peer-reviewed and formally published. If this work was previously peer-reviewed and published, in the cover letter please provide the reason that this work does not constitute dual publication and should be included in the current manuscript.

Reviewers' comments:

Reviewer's Responses to Questions

**Comments to the Author**

1. Is the manuscript technically sound, and do the data support the conclusions?

Reviewer #1: Partly

Reviewer #2: Yes

2. Has the statistical analysis been performed appropriately and rigorously? 

Reviewer #1: No

Reviewer #2: No

3. Have the authors made all data underlying the findings in their manuscript fully available?

Reviewer #1: Yes

Reviewer #2: Yes

4. Is the manuscript presented in an intelligible fashion and written in standard English?

Reviewer #1: No

Reviewer #2: Yes

5. Review Comments to the Author

Reviewer #1: Comments:

I have few comments for this manuscript given in each section, as follows:

Introduction

1. In the first sentence of the third paragraph, “youths aged 15 to 24 years were more likely to report STI and its symptoms than older men”. Do “youths” in this sentence specifically refer to male youths? This is due to the comparison group is older men. Please clarify.

2. The second sentence in the same paragraph, “the prevalence of HIV/AIDS …. was 11.5%” is vague. Please clarify whether the figure of 11.5% represents the prevalence of HIV or the prevalence of AIDS. In addition, 11.5% was considerably high prevalence for young people aged 20-24 years. Please check again the source and/or interpretation of the data.

3. In many settings, adolescents (10-19 years) are also vulnerable to HIV infection. The prevalence of protected sex among this group tends to be low due to insufficient HIV knowledge and facing more barriers in accessing contraceptive methods (e.g. condom). Please provides strong reasons why this study focused on male college youths (19-24 years) instead of males at secondary educational level (10-19 years).

Methods

1. The information on data analysis is missing. Please add information on how data were analysed in the methods section.

2. What are the validity and reliability of the condom knowledge and attitude towards condom? Please elaborate more.

Results

1. Even though this paper only analysed the data from 361 students due to eligibility criteria of vaginal or anal sex, it is better to inform the readers about the characteristics of all recruited students (903). Authors can add a cross-tabulation table between characteristics and sexual activity (anal and/or vaginal sex) along with the results of Chi-square test as a supplementary file or an appendix. This potentially informs the readers what students’ characteristics prone to have vaginal and/or anal sex. Authors can put interpretation for this table in the results section and no need to explain in detail since it is not the aim of this study.

2. Before summing up the total score of knowledge and attitude and grouping into binary variables, it is important to have a table listing indicators used to measure knowledge and attitude and present the proportions of students who answered correctly for each item. This can inform authors and readers what components of knowledge and attitude are still lacking among respondents and potentially inform for policy recommendations.

3. Even though there was multicollinearity among independent variables, the bivariate analyses should be presented for all independent variables. Therefore, please add the bivariate associations for age and condom use; level of education and condom use. In addition, please elaborate more on multicollinearity testing informed in the results section. Were variables of age and level of education omitted due to a high correlation between both?

4. What does the symbol “#” mean in p-value of the Chi-square test for marital status and condom use?

5. Please be consistent in presenting the information for a continuous variable, such as age at first sex that can follow the information presented for condom knowledge (mean ±SD: ; Min-Max).

6. Please be consistent in grouping a variable of age at first sex in the descriptive table (Table 3) and in the table of bivariate and multivariate analyses (Table 4). The same variable was grouped in different ways. This does not make sense.

7. For presenting the cross-tabulation (Table 4), please use row percentage, not column percentage. Please revise the table accordingly.

8. There are a lot of mistakes in Table 4. I could not find where is the OR for the association between the sexual partner at last sex and condom use. However, authors interpreted the results of this association.

Discussion

1. The discussion section is lacking information on how associations between independent variables and condom use occurred in the context of Nepal. For example, those with unfavourable attitude towards condom were less likely to use a condom at the last sex, but no adequate explanations for this. Please explain why this study found this association within a sample of Nepalese youths. What does “unfavourable attitude” mean? Please elaborate more on other associations too.

2. “In this study, 60.4% had two or more sex partners, which is in line with the findings of Kathmandu, Nepal, where 54.9% had so”. This sentence is not clear. Please revise and add a reference if authors want to compare to previous study findings.

3. Please elaborate more in one paragraph about the implication of the study findings. What can authors suggest to increase the use of condoms among male youths by considering socio-cultural norms as barriers to condom use among unmarried youths?

4. In addition to study limitations, please add the strengths of this study.

Addictional comments:

The manuscript should be proofread by a native speaker before further submission.

Reviewer #2: Review opinion.

Although sex life is a difficult theme to research, no matter how anonymous it is, good results have been obtained through a detail research design. Research about the use of condom in youth and the impact on young people’s life has been meaningful. However, the following corrections are required.

1. Research Design

The process of subject selection for the study needs to be further explained; e.g.: the study population, the number of students per class, how did you choose the classes (randomly or not), and the student’s major.

2. Knowledge and attitude toward the use of condoms

Please explain why you used the mean score. In particular, knowledge is measured in a 5-point scale, but there is no explanation on the basis for dividing the results into appropriate or inappropriate.

Also, the variable attitude presents the same issue.

3. Table 4. Needs to be organized and corrected for typos.

4. Discussion

- Considering that this study was conducted on university and graduate students aged 19-24, the results of the study need to be compared with prior studies that also consider the age of the subjects.

- The discussion of the results seems very weak when compared with prior studies.

- Logic is very weak when discussing number of partners and condom use. Consider reviewing related prior studies.

- There is no supporting information explaining the use of condom for married people.

6. PLOS authors have the option to publish the peer review history of their article (what does this mean?). If published, this will include your full peer review and any attached files.

Reviewer #1: No

Reviewer #2: **Yes: **Eun Woo Nam

---

## [Author Response · Author response to Decision Letter 0]

23 Jan 2021

Reviewer 1 and 2: We have incorporated all of your suggestions in the revision except the weighted calculations. We calculated the weighted estimator of the sample and found 1.04 when the formula Wi=K/Si (Tyagi, S., Sharma, B., Singh, P. and Dobhal, R., 2013. Water quality assessment in terms of water quality index. american Journal of water resources, 1(3), pp.34-38.) was applied. Hence the estimator is minimal, it does not make a large effect, we did not feel necessary to carry out further calculations as implicitly mentioned by Korn and Graubard(Korn, E.L. and Graubard, B.I., 1995. Examples of differing weighted and unweighted estimates from a sample survey. The American Statistician, 49(3), pp.291-295.). 

We would like to thank both reviewers since your comments helped to refine the manuscript to a great extent. Although we believe the manuscript has reached to an acceptable level, we still appraise truly if further comments and feedback deemed necessary.

---

## [Decision Letter · Decision Letter 1]

18 Feb 2021

PONE-D-20-30105R1

Determinants of condom use during last sexual intercourse among male college youths of Kaski, Nepal: A cross-sectional survey

PLOS ONE

Dear Dr. Adhikari,

Thank you for submitting your manuscript to PLOS ONE. After careful consideration, we feel that it has merit but does not fully meet PLOS ONE’s publication criteria as it currently stands. Therefore, we invite you to submit a revised version of the manuscript that addresses the points raised during the review process.

The manuscript has improved but it still requires important editing. Particularly in the description of methods and analysis, as suggested by the reviewer, and language. There is still a need to copy edit, and to tone down some of the claims. Particularly the introduction NEEDS an improvement. I am pointing out incorrections, but the whole thing needs to be improved. If there was an English language editing, the editor did a very bad job. There are still many incorrections and the article cannot be published unless the language improves drastically You can do a grammar check with any word processor or grammarly.com. There are plenty of problems.  Also, for future revisions, make sure that the line numbers are included in the manuscript with track changes and that the detailed answers to the suggestions are submitted together with the revised manuscript.

A first required change is the use of “male college youths” in some parts of the text instead of “male college youth”, which is also used. Youth can be plural, and use of “youths” should be avoided. Please replace all uses of “youths” for “youth”.“Among the various health issues related to sexual and reproductive health, the major concern of youths is that they expose to the Sexually Transmitted Infections”. Please rephrase. The sentence does not make grammatical sense. Who is “they”?. Also, use “a major concern”, it is unclear that it is “the” major concern, and that does not stem from the reference given.“There is the variation of sexual activities of youths between the countries of the world”: It should be “there is variation in sexual activity of youth among ..“But the common problem faced by most of the countries is that young people are reaching puberty earlier and are often engaged in sexual activities”. Countries do not face problems, and it is debatable whether reaching puberty earlier and engaging in sexual activities is a problem. You have to use neutral language. The same for the following sentence “. You could write something as “A common pattern is for youth reaching puberty and engaging in sexual activity earlier. Evidence shows earlier sexual initiation to be associated with having multiple sex partners  and higher risk of sexually transmited infections”.You should introduce at this stage evidence from the prior study in JHAS such as proportion ever sexually active and multiple partnership in Kaski compared to DHS or the other studies in Achham and Kathmandu.“These existing literature of Nepal indicates that the sexual behavior of youths of Nepal are risky.” Revise. It does not make grammatical sense.You also need to refer to the other article in the statement of the research objectives. You just say “Despite the efforts from various level, the prevalence of condom use among youths is low [12]. So this study examined the condom use at last sexual intercourse among the male college youths.”. You are not looking at this in the vacuum. You are looking at youth in a particular city where you have studied sexual behaviour. This is important as commented before: “If a submitted study replicates or is very similar to previous work, authors must provide a sound scientific rationale for the submitted work and clearly reference and discuss the existing literature. Submissions that replicate or are derivative of existing work will likely be rejected if authors do not provide adequate justification (http://www.plosone.org/static/publication#results). The mentioned paper was not included among the reference or put in context as required, and if the required justification and incorporation in the study is not given that is grounds for rejection”. This justification needs to be incorporated in the text as an antecedent of the current study. Also you are saying in the response that this study uses “new data”. You have to clarify that: Was there one interview which is common to both studies or two different interviews? This is confusing.“It is evident that several factors influences condom use among youths as their sociodemographic characteristics: residence, education, living arrangement”. Nothing is evident. That is not, in particular, evident. Also, verb and noun should agree in number. The English language of the article really needs improvement.Sample weights are still not used. The description of sampling has been provided: “The total number of classes of undergraduate level as: Bachelor first ,second, third and fourth and year was 87,86.84 and 58 respectively and of post graduate was 21.As per the population proportion to size the total number of classes to be taken from undergraduate to post graduate level was 46 .Thus the 46 classes were selected by generating random number and all male college youths of the selected classes were the study population.”. First, language should still be improved. Second, it emerges that there are unequal probabilities of selection. Students in smaller classes have a higher probability of selection since there is a random sample of classes, not of students. The argument given for not using weights is a reference to an article on water management that is not relevant methodologically, and the reproduced equation does not make sense in this setting. You are not calculating a weighted average. You are estimating a GLM model with unequal probabilities of selection. This requires use of normalized weights with an average level of one.The section on the results of the multivariate analysis reads very poorly. It should be a little more than reproducing the elements of the table. Think of the reader.Discussion: Needs improvement in the language and the substance. For instance, what does it mean “The study revealed the aspects of youth's sexual behavior that may point to advance the well-being of youth's reproductive and sexual health”. Besides the clumsy redaction that does not seem to be the purpose of the article. You have looked at the determinants of condom use at last intercourse, and there are no policy variables included.

We look forward to receiving your revised manuscript.

Kind regards,

José Antonio Ortega, Ph.D.

Academic Editor

PLOS ONE

Reviewers' comments:

Reviewer's Responses to Questions

**Comments to the Author**

1. If the authors have adequately addressed your comments raised in a previous round of review and you feel that this manuscript is now acceptable for publication, you may indicate that here to bypass the “Comments to the Author” section, enter your conflict of interest statement in the “Confidential to Editor” section, and submit your "Accept" recommendation.

Reviewer #1: (No Response)

2. Is the manuscript technically sound, and do the data support the conclusions?

Reviewer #1: Yes

3. Has the statistical analysis been performed appropriately and rigorously? 

Reviewer #1: No

4. Have the authors made all data underlying the findings in their manuscript fully available?

Reviewer #1: Yes

5. Is the manuscript presented in an intelligible fashion and written in standard English?

Reviewer #1: No

6. Review Comments to the Author

Reviewer #1: Revisions for the given comments were satisfactory. However, I have some comments that need to be addressed by the authors, as follows:

1. Pokhara and Kaski are interchangeably used in the manuscript, such as in the last sentence in the introduction and some parts of the results. Do both names refer to the same district?

2. In the sub-section of questionnaire, please provide the citations from where questionnaires were adopted.

3. In the methods section, please make clear whether any records with missing data that were removed from 361 males who engaged in sexual intercourse in last 6 months.

4. For bivariate analysis (Table 6), please mention the reference category or group for all variables. No reference group was mentioned for the age of the respondent and educational level.

5. For the same table, please check again the association between the number of sex partners and condom use. From the table, those who had 2 partners or more were more likely to use a condom at last sex (OR=8.91; 95%CI=5.21-15.24). Surprisingly, based on the differences in the proportion of condom use by the number of sex partners, the proportion of condom use was lower among those with ≥ 2 partners (11.01%) compared to those with 1 partner only (52.44%). Please check again the results whether there was a mistake in coding the categories. Please check also for other variables: level of education, relationship with sex partners, and provide interpretation for Table 6.

6. Authors reported a high correlation between the age of the respondents and level of education (r=0.9) and decided to not include both variables in the multivariate model. However, two highly correlated variables indicate that both cannot be included in the multivariate model in a time, but we can select one of them. Therefore, authors can include either age of the respondents or educational level in the model. I suspect that educational level can be strongly associated with condom use.

7. For multivariate analysis (Table 7), I am wondering why the number of sex partners was not statistically significant associated with condom use. The 95%CI (3.01 – 11.07) informs that the value of one (1) is not within the intervals. Please re-check the results.

7. PLOS authors have the option to publish the peer review history of their article (what does this mean?). If published, this will include your full peer review and any attached files.

Reviewer #1: No

---

## [Author Response · Author response to Decision Letter 1]

26 Jul 2021

We are very much thankful to the editor and the reviewers for your generous comments on the manuscript entitled " Determinants of condom use during last sexual intercourse among male college youths of Kaski, Nepal: A cross-sectional survey". We hope the scientific comments you aroused have helped rectify the manuscript and accordingly, we have revised it by addressing all the concerns raised by editor and reviewers. The responses along with the comments are described in this rebuttal letter. We also apologize for a bit delayed response. 

Editor comments and changes /justification

1. Comment # 1: A first required change is the use of “male college youths” in some parts of the text instead of “male college youth”, which is also used. Youth can be plural, and use of “youths” should be avoided. Please replace all uses of “youths” for “youth”.

Our response: We thank the editor for this lexical error and all "youths” are now replaced by “youth” in the manuscript.

2. Comment # 2: Among the various health issues related to sexual and reproductive health, the major concern of youth is that they expose to the Sexually Transmitted Infections”. Please rephrase. The sentence does not make grammatical sense. Who is “they”?.Also, use “a major concern”, it is unclear that it is “the” major concern, and that does not stem from the reference given.

Our Response: Thank you for this lexical and grammatical issue. We have, now, rephrased it as: “The youth face various health problems that can affect their quality of life. First and foremost is HIV/AIDS, which is increasingly affecting young people.” We hope the sentence is sensible now.

3. Comment # 3: But the common problem faced by most of the countries is that young people are reaching puberty earlier and are often engaged in sexual activities”. Countries do not face problems, and it is debatable whether reaching puberty earlier and engaging in sexual activities is a problem. You have to use neutral language. The same for the following sentence “. You could write something as “A common pattern is for youth reaching puberty and engaging in sexual activity earlier.

Our response: Thank you again for notifying us. The sentence "But the common problem faced by most of the countries is that young people are reaching puberty earlier and are often engaged in sexual activities" is now written as "A common pattern is for youth reaching puberty and engaging in sexual activity earlier".

4. Comment # 4: Evidence shows earlier sexual initiation to be associated with having multiple sex partners and higher risk of sexually transmited infections”. You should introduce at this stage evidence from the prior study in JHAS such as proportion ever sexually active and multiple partnership in Kaski compared to DHS or the other studies in Achham and Kathmandu.

Our response: We are thankful to the generous comment. Prior study of JHAS ,Achham ,Kathmandu and DHS data are now added and arranged with this sentence. We found it as improving the rigor of the manuscript. 

5. Comment # 5: These existing literature of Nepal indicates that the sexual behavior of youths of Nepal are risky.” Revise. It does not make grammatical sense.

Our response: The sentence is now revised as " Thus, the literature suggests that the sexual activity of youth in Nepal is risky." 

6. Comment # 6: You also need to refer to the other article in the statement of the research objectives. You just say “Despite the efforts from various level, the prevalence of condom use among youths is low [12]. So this study examined the condom use at last sexual intercourse among the male college youths.” You are not looking at this in the vacuum. You are looking at youth in a particular city where you have studied sexual behaviour. This is important as commented before.

Our response: We are thankful for the comments. More literatures for the justification have been added in the revised manuscript

7. Comment # 7: If a submitted study replicates or is very similar to previous work, authors must provide a sound scientific rationale for the submitted work and clearly reference and discuss the existing literature. Submissions that replicate or are derivative of existing work will likely be rejected if authors do not provide adequate justification (http://www.plosone.org/static/publication#results). The mentioned paper was not included among the reference or put in context as required, and if the required justification and incorporation in the study is not given that is grounds for rejection”. This justification needs to be incorporated in the text as an antecedent of the current study. Also you are saying in the response that this study uses “new data”. You have to clarify that: Was there one interview which is common to both studies or two different interviews? This is confusing.

Our response: In line with the editor’s comment, we have now introduced our prior study of JHAS in the sentence "Evidence shows earlier sexual initiation to be associated with having multiple sex partners and higher risk of sexually transmitted infections” ,and the mentioned paper is also included in the reference in the 8th number. In addition, the mentioned paper study is also discussed in the context of multiple sex partners at the discussion section.

• It is also explained as the rationale in the last paragraph of introduction that "Previous study conducted among male college youth of Kaski district revealed that the substantial proportion of male college youth was engaged in risky sexual behavior where more than half (51.61%) of the male youth had not used condom at an every act of sexual intercourse with commercial sex worker. So this study is important to examine the prevalence and determinants of condom use at last sexual intercourse among male college youth of Kaski district"

• Regarding the interview, two different interviews were conducted. Prior study (published in JHAS) among the participants who were sexually active during the last one year period, after preliminary findings of which further intrigued us to think about the condom use during the last vaginal/anal intercourse (within 6 months) would be even more interesting and important from the scientific point of view, and so accessed among those and included as per the criteria, which helped us to get to 361. This study is, so, the further culmination of previous study which, probably, was able to answer, another but an important research question. 

8. Comment # 8: It is evident that several factors influence the condom use among youth as their sociodemographic characteristics: residence, education, living arrangement”. Nothing is evident. That is not, in particular, evident. Also, verb and noun should agree in number. The English language of the article really needs improvement.

Our response: It is revised as "Studies have documented that living with parents and higher education level are the important predictors of consistent condom use"

9. Comment # 9: Sample weights are still not used. The description of sampling has been provided: “The total number of classes of undergraduate level as: Bachelor first ,second, third and fourth and year was 87,86.84 and 58 respectively and of post graduate was 21.As per the population proportion to size the total number of classes to be taken from undergraduate to post graduate level was 46 .Thus the 46 classes were selected by generating random number and all male college youths of the selected classes were the study population.”. First, language should still be improved. Second, it emerges that there are unequal probabilities of selection. Students in smaller classes have a higher probability of selection since there is a random sample of classes, not of students. The argument given for not using weights is a reference to an article on water management that is not relevant methodologically, and the reproduced equation does not make sense in this setting. You are not calculating a weighted average. You are estimating a GLM model with unequal probabilities of selection. This requires use of normalized weights with an average level of one.

Further commented/clarification on 16th July, 2021 (upon our request): Regarding the question of Dr. Adhikari, please reply to him that I'd be glad to see their resubmission. Turning to their question regarding point 9:

Upon re-reading of the sampling procedure, it seems that there might be (prior) equal probabilities of selection since all male students in each sampled class were targeted so the probability of selection is given by the proportion of classes surveyed to all classes. There would be unequal probabilities if a fixed number of students were sampled, but this is not apparently the case. However, ex-post there are still different probabilities of sampling in each of the education levels. Eg: What if all the 46 randomly selected classes belonged to the 87 first-year classes? The prior probability of choosing students from each level is the same as in the population, but, after the sample is taken, the proportion of students from each level in the sample could be as drastic as only first-year students. Note you could use weights to adjust for this (unless one education level is totally missing!) defined as the proportion of level i (eg: first year students) in the population divided by the proportion of first year students in your sample. In this way you give more weight to students from levels that are underrepresented and the weights are normalized (have an average of one). This is equivalent to using strata. You can use these weights (defined at the student-year level, first to fourth and postgraduate) in your analysis. Alternatively, you can show that your sample is balanced, (eg, if all weights calculated in this way are very close to one) and not use weights.

Still, the paragraph was hard to read. Eg: "and all male college youths of the selected classes were the study population". This confuses sample and population. All the male college youths in the selected classes constitute your sample. The population includes all male college youths, in all classes irrespective of sampling status.

Second, the sentence "As per the population proportion to size the total number of classes to be taken from undergraduate to post graduate level was 46" is also confusing". Do you mean something like "based on average classroom size, 46 classes would have to be targeted to reach desired sample size". If this is so, then put the paragraph that follows on targeted sample size before and not after describing the sample selection. Note that given unequal classroom sizes what varies depending on the classroom sample is sample size, which is not fixed.

In short: there is a need for a thorough language check. Regarding weights, you can either show that your sample is balanced and the proportions in the sample of students by year are similar to the proportions in the population, or else use weights as suggested (ratio of the previously mentioned proportions).

Our response: We are very much thankful to your comments and the language feedback. Firstly, regarding the thorough language check, we took it seriously, and all three authors, checked and peer reviewed the language after corrected in online Grammerly. that was checked after revising the manuscript as commented and feedback. The illogical, insensible and unintelligible sentences that have been commented have been revised in consultation with the professional language expert. 

Secondly, and may be, even more importantly, in response to the weightage calculation/balanced sample, we are grateful to the reviewers and editors, for providing the academic suggestions. As per your suggestions and to be more assured, we consulted the biostatistician and his verdict was consistent with your later one. Henceforth, we calculated the weighted proportion of the actual male youth and compared it with the unweighted number that was sampled for study, and it revealed that the sample is balanced. We also carried out the chi-square test if there exists any association between the number of students of weighted and unweighted and the finding was insignificant (chi-square statistic, 0.60; p-value, 0.96) and concluding that there was no association between the two data sets (supporting file 5).

Thirdly, the doubt that has been aroused by the reviewer that the disproportionate sampling would have occur and all the classrooms would have been selected only from bachelor level, we were conscious about that, and we had then selected the classrooms from each of bachelor’s 1st, 2nd, 3rd, 4th, and master’s 1st years in accordance with the proportion to the number of students in the classrooms. So, the representation from all the levels was assured (supporting file 5).

10. Comment # 10: The section on the results of the multivariate analysis reads very poorly. It should be a little more than reproducing the elements of the table. Think of the reader.

Our response: Thinking the reader, as per your suggestions, the manuscript in now revised and added the explanation of each model so that author can connect the interpretation with table easily.

11. Comment # 11: Discussion: Needs improvement in the language and the substance. For instance, what does it mean “The study revealed the aspects of youth's sexual behavior that may point to advance the well-being of youth's reproductive and sexual health”.Besides the clumsy redaction that does not seem to be the purpose of the article. You have looked at the determinants of condom use at last intercourse, and there are no policy variables included.

Our response: “The study revealed the aspects of youth's sexual behavior that may point to advance the well-being of youth's reproductive and sexual health” is replaced by The study examined the determinants of condom use during last sexual intercourse among sexually active male youth of Kaski, Nepal. We have also thoroughly checked the manuscript, that has already clarified in response of comment 9. 

As the study had focused on determinants of condom use during last sexual intercourse and had not included policy variables so policy related aspects are no more discussed now in the revised manuscript and is omitted.

12. Upon re-reading of the sampling procedure, it seems that there might be (prior) equal probabilities of selection since all male students in each sampled class were targeted so the probability of selection is given by the proportion of classes surveyed to all classes. There would be unequal probabilities if a fixed number of students were sampled, but this is not apparently the case. However, ex-post there are still different probabilities of sampling in each of the education levels. Eg: What if all the 46 randomly selected classes belonged to the 87 first-year classes? The prior probability of choosing students from each level is the same as in the population, but, after the sample is taken, the proportion of students from each level in the sample could be as drastic as only first-year students. Note you could use weights to adjust for this (unless one education level is totally missing!) defined as the proportion of level i (eg: first year students) in the population divided by the proportion of first year students in your sample. In this way you give more weight to students from levels that are underrepresented and the weights are normalized (have an average of one). This is equivalent to using strata. You can use these weights (defined at the student-year level, first to fourth and postgraduate) in your analysis. Alternatively, you can show that your sample is balanced, (eg, if all weights calculated in this way are very close to one) and not use weights.

Our response: This comment was given on 16th Jul, 2021 and included in comment 9, above, and responded accordingly. 

Comments of Reviewer 1 and changes/justification

1. Pokhara and Kaski are interchangeably used in the manuscript, such as in the last sentence in the introduction and some parts of the results. Do both names refer to the same district?

Our response: We are thankful to your comments. Consistency has been maintained through out the manuscript by using the word Kaski.

2. In the sub-section of questionnaire, please provide the citations from where questionnaires were adopted.

Our response: Dear reviewer, citations are now provided while explaining the questionnaires.

3. In the methods section, please make clear whether any records with missing data that were removed from 361 males who engaged in sexual intercourse in last 6 months.

Our response: Dear reviewer, it is now explained in the statistical method section of manuscript and written as Altogether there were 74 ambiguous responses of the sexually active male youth in which 12 responses were of sexually active male youth within six months preceding the survey. Likewise, among 21 incomplete responses, 6 largely incomplete responses were of sexually active male youth within six months preceding the survey. Thus, ambiguous responses that were logically inconsistent were excluded after verification through the consultation of reproductive health experts and incomplete data were also eliminated from the analysis.

4. For bivariate analysis (Table 6), please mention the reference category or group for all variables. No reference group was mentioned for the age of the respondent and educational level.

Our response: The reference group is now mentioned for the age of the respondent and educational level.

5. For the same table, please check again the association between the number of sex partners and condom use. From the table, those who had 2 partners or more were more likely to use a condom at last sex (OR=8.91; 95%CI=5.21-15.24). Surprisingly, based on the differences in the proportion of condom use by the number of sex partners, the proportion of condom use was lower among those with ≥ 2 partners (11.01%) compared to those with 1 partner only (52.44%). Please check again the results whether there was a mistake in coding the categories. Please check also for other variables: level of education, relationship with sex partners, and provide interpretation for Table 6.

Our response: The result of all variables of Table 6 was checked and it was found that there was mistake on data for the association between the number of sex partners and condom use. The proportion of condom users among ≥ 2 partners is 194 (89 %) and with 1 partner only is 68(47.5%).For other variable also it was checked and no mistake was found.

6. Authors reported a high correlation between the age of the respondents and level of education (r=0.9) and decided to not include both variables in the multivariate model. However, two highly correlated variables indicate that both cannot be included in the multivariate model in a time, but we can select one of them. Therefore, authors can include either age of the respondents or educational level in the model. I suspect that educational level can be strongly associated with condom use.

Our response: Educational level is now added in the model.

7. For multivariate analysis (Table 7), I am wondering why the number of sex partners was not statistically significant associated with condom use. The 95%CI (3.01 – 11.07) informs that the value of one (1) is not within the intervals. Please re-check the results.

Our response: The result was checked and it was found that sex partners was statistically significant with condom use in table 7 so the manuscript is revised as per it.

---

## [Editor Report · Decision Letter 2]

6 Dec 2021

Determinants of condom use during last sexual intercourse among male college youths of Kaski, Nepal: A cross-sectional survey

PONE-D-20-30105R2

Dear Dr. Adhikari,

We’re pleased to inform you that your manuscript has been judged scientifically suitable for publication and will be formally accepted for publication once it meets all outstanding technical requirements.

Kind regards,

José Antonio Ortega, Ph.D.

Academic Editor

PLOS ONE

Additional Editor Comments (optional):

The editor wants to commend the authors for a manuscript very much improved that now reads profesionally and has overcome the main limitations of analysis and interpretation making it amenable for publication. It is not felt necessary to send the manuscript back to the reviewers considering the appropriate reaction to reviewer 1 comments and that it was the editor who had the strongest reservations on earlier drafts.
---

## [Editor Report · Acceptance letter]

17 Dec 2021

PONE-D-20-30105R2 

Determinants of condom use during last sexual intercourse among male college youth of Kaski, Nepal: A cross-sectional survey 

Dear Dr. Adhikari:

I'm pleased to inform you that your manuscript has been deemed suitable for publication in PLOS ONE. Congratulations! Your manuscript is now with our production department. 

Kind regards, 

on behalf of

Dr. José Antonio Ortega 

Academic Editor

PLOS ONE